# Molecular Mechanism of Action of RORγt Agonists and Inverse Agonists: Insights from Molecular Dynamics Simulation

**DOI:** 10.3390/molecules23123181

**Published:** 2018-12-03

**Authors:** Nannan Sun, Congmin Yuan, Xiaojun Ma, Yonghui Wang, Xianfeng Gu, Wei Fu

**Affiliations:** Department of Medicinal Chemistry and Key Laboratory of Smart Drug Delivery, Ministry of Education, School of Pharmacy, Fudan University, Shanghai 201203, China; nysnn@126.com (N.S.); 16211030013@fudan.edu.cn (C.Y.); scuma@foxmail.com (X.M.); yonghuiwang@fudan.edu.cn (Y.W.)

**Keywords:** RORγt, molecular mechanism of action (MOA), agonist, inverse agonist

## Abstract

As an attractive drug-target, retinoic acid receptor-related orphan receptor-gamma-t (RORγt) has been employed widely to develop clinically relevant small molecular modulators as potent therapy for autoimmune disease and cancer, but its molecular mechanism of action (MOA) remains unclear. In the present study, we designed and discovered two novel RORγt ligands that are similar in structure, but different in efficacy. Using fluorescence resonance energy transfer (FRET) assay, compound **1** was identified as an agonist with an EC_50_ of 3.7 μM (max. act.: 78%), while compound **2** as an inverse agonist with an IC_50_ value of 2.0 μM (max. inh.: 61%). We performed molecular dynamics (MD) simulations, and elucidated the MOA of RORγt agonist and inverse agonist. Through the analyses of our MD results, we found that, after RORγt is bound with the agonist **1**, the side chain of Trp317 stays in the *gauche*- conformation, and thus helps to form the hydrogen bond, His479-Trp502, and a large hydrophobic network among H11, H11′, and H12. All these interactions stabilize the H12, and helps the receptor recruit the coactivator. When the RORγt is bound with the inverse agonist **2**, the side chain of Trp317 is forced to adopt the *trans* conformation, and these presumed interactions are partially destroyed. Taken together, the critical role of residue Trp317 could be viewed as the driving force for the activation of RORγt.

## 1. Introduction

The nuclear receptor (NR) retinoic acid receptor-related orphan receptor-gamma-t (RORγt, also known as NR1F3) is an important transcription factor involved in the differentiation of T helper 17 (Th17) cell and production of the pro-inflammatory cytokine, interleukin 17 (IL-17) [1]. The recent success of IL-17 antibodies (Secukinumab, Ixekizumab, and Brodalumab) and the progress of RORγt inverse agonists in clinical trials (VTP-43742, GSK2981278A, ARN-6039, TAK-828, ABBV-553, JNJ-3534, AZD-0284, JTE-451, JTE-151, RTA-1701) established that RORγt is a valuable drug-target for the treatment of autoimmune diseases [2,3,4,5,6,7,8]. Recent research has revealed that RORγt plays a critical role in the generation and function of Th17 and cytotoxic T (Tc17) cells [9,10,11,12,13]. With the clinical proof of concept inferred from a small molecule RORγt agonist, LYC-55716, for the treatment of cancer, it is also proposed that RORγt is a potential target to develop small molecular therapeutics against certain types of cancer [14].

As a typical NR, RORγt contains a variable N-terminal domain (A/B or AF1), a conserved DNA-binding domain (DBD), a flexible hinge region, and a C-terminal ligand-binding domain (LBD) with the ligand-dependent activation function helix (AF2). The RORγt-LBD comprises the 12 canonical α-helices (H1-H12) along with two additional helices (H2′ and H11′). Small molecules could bind to the LBD to modulate the gene transcription function of RORγt [9,15,16]. As shown in Figure 1A, RORγt agonists enhance gene transcription by stabilizing H12 and promoting the recruitment of a coactivator. On the contrary, RORγt inverse agonists disarray H12, which compromise the recruitment of coactivator hard yet promote the recruitment of a corepressor, and generally decreases gene transcription (Figure 1B). It is valuable to explore the molecular mechanism of action (MOA) of RORγt agonists and inverse agonists to guide the design of more potent small molecule therapeutics for cancer and autoimmune disease.

Considerable efforts have been directed toward the discovery of RORγt agonists and inverse agonists [17,18,19,20]. There is an interesting phenomenon in these explorations, where a minor change on the structure of some RORγt ligands may lead to distinct MOA (Figure 2), such as the tertiary sulfonamides, biaryl amides, tertiary amines, benzoxazinones, etc. [21,22,23,24,25]. How do the molecular changes induce different MOA of RORγt? Till now, 67 crystal structures of RORγt have been deposited in the Protein Data Bank (PDB, https://www.rcsb.org). Comparison of multiple co-crystal structures of RORγt with agonists and inverse agonists is helpful to understand the agonism and inverse agonism conformation of RORγt to a certain extent. Overlay of co-crystal structures with agonists and inverse agonists in the orthostatic binding site reveals that H11, H11′, and H12 diverge slightly. How could a minor structure change of the ligand lever the secondary structure of H11′ and H12?

In the present study, to explore the MOA of RORγt agonists and inverse agonists, we designed and synthesized two potent RORγt ligands with a novel *N*-sulfonamide tetrahydroquinoline scaffold, a potent agonist **1** and inverse agonist **2** that were identified by assay of fluorescence resonance energy transfer (FRET). The following molecular dynamics (MD) simulations identified the key role of two rotamers of residue Trp317 in the stabilization of H11′ and H12, and the MOA of RORγt’s agonist and inverse agonist is elucidated at the molecular level.

## 2. Methods and Materials

### 2.1. Chemical Synthesis

The new *N*-sulfonamide tetrahydroquinoline derivatives were synthesized by following the synthetic approach (Scheme 1). 6-Bromo-3,4-dihydroquinolin-2(1*H*)-one was reduced to generate 6-bromo-1,2,3,4-tetrahydroquinoline. 6-bromo-1,2,3,4-tetrahydroquinoline was reacted with benzene sulfonyl chloride/benzylsulfonyl chloride and triethylamine in dichloromethane (DCM) at room temperature. Then, we coupled the bromo-substituent sulfamide intermediates to 1-(piperazin-1-yl) ethan-1-one by a Buchwald-Hartwig coupling reaction in the presence of Palladium(II) acetate, 2-dicyclohexylphosphino-2′,6′-di-i-porpoxy-1, and 1′-biphenyl and Cs_2_CO_3_. In this way, we obtained sulfonamide derivatives **1** and **2**, which have only a carbon change in structure. Data of structural identification is presented in the Appendix A.

### 2.2. RORγt Based dual FRET Assay

We performed a fluorescence resonance energy transfer (FRET) assay (details are provided in the Appendix A) to measure biological activities of compounds in human RORγt-LBD, in the presence of a coactivator peptide, which was derived from steroid receptor coactivator (SRC)-1. 

### 2.3. Mouse Cell Th17 Differentiation Assay

We further evaluated the two novel compounds in a mouse Th17 cell differentiation assay (details are provided in the Appendix A) to test the activity on the transcription of IL-17. 

### 2.4. Molecular Docking

To explore how the agonist **1** and inverse agonist **2** bind with the RORγt-LBD, we performed molecular docking by using the Schrodinger 3.5 software package. The co-crystal structure of RORγt-LBD (PDB code: 4NIE, 5APK and 5VB3) were extracted directly from the protein data bank, and processed by using the Protein Preparation Wizard, including water deletion, addition of missing hydrogen atoms as well as adjusting of the tautomerization and protonation states of histidine. The 3D structures of compounds were optimized by energy minimizations with force field (OPLS_2005) before submitting to the docking procedure. The possible binding site for agonist **1** was assumed to be similar as that was reported in the agonist-bound co-crystal structure (PDB ID: 4NIE), and the binding site for inverse agonist **2** should be similar as that in the inverse agonism system (PDB ID: 5APK). The docking grid was set at the center of the ligand structure, and the side length of the bounding box was set to 15 Å. The docking was performed with Glide-docking using the Extra Precision (GlideXP) algorithm. The final ranking from the docking was based on the docking score, which combines the Epik state penalty with the Glide Score.

### 2.5. System Preparation for MD

The apo, agonistic, and inverse agonistic systems were set up for MD simulations. Until now, no apo crystal structure of RORγt-LBD was reported, but a chimeric protein: His6-RORγt-LBD (260–507)-GGG-EKHKILHRLLQDS (SRC2 peptide) [26]. The structure of an apo RORγt-LBD was constructed by deleting the tri-glycine linker and SRC2 of the solved chimeric protein by means of the Maestro program. The docked complexes of agonist **1**-RORγt-LBD (PDB: 4NIE) and inverse agonist **2**-RORγt-LBD (PDB: 5APK) were used as initial structures of MD simulations for agonistic and inverse agonistic systems in this study. The topologies of agonist **1** and inverse agonist **2** were generated from the PRODRG server (http://davapc1.bioch.dundee.ac.uk/cgi-bin/prodrg/submit.html). 

### 2.6. MD Simulations

Simulations of the three systems were performed using a Gromacs 5.1.4 package applied with the GROMOS96 43A1 force field [27]. These systems were enclosed in the SPC cubic water box with the protein atoms located 10 Å between the protein surface and the box boundary. Each of the three systems was neutralized by adding Na^+^ and Cl^−^ ions at the ionic concentration of 0.15 M. The periodic boundary condition (PBC) was employed in all directions of the simulation box. To remove unfavorable steric clashes in the protein structures, energy minimization was firstly carried out using the steepest descent algorithm followed by the unrestrained conjugate gradient algorithm. Then, the energy minimized structure was equilibrated with position-restrained MD simulations in the NVT ensemble: A constant number of particles, volume, and temperature, with a time step of 2 fs. The simulation system was heated to 300 K in 35 ps gradually with the aid of a Langevin thermostat. The MD simulations were performed with the output structure from the position-restrained dynamics simulations at 100 ps in the NPT ensemble at 300 K. The Parrinello-Rahman method was employed to maintain a constant pressure of 1 atm. The LINCS algorithm was used to constrain all bounds [28] and the Van der Waals interaction cutoff was set to be 10 Å while long-range electrostatic interactions were calculated using Particle Mesh Ewald (PME). Finally, the production MD simulations of 100 ns were completed for each system. The MD trajectories were analyzed using tools implemented inside the Gromacs package. Visualization and inspection of the trajectories were performed with the VMD software (https://www.ks.uiuc.edu/Research/vmd/). 

## 3. Design, Results, and Discussion

RORγt agonist can activate the receptor and make it be able to recruit coactivator by stabilizing H12, and then the activated receptor will enhance the gene transcription. The RORγt inverse agonist is suggested to destabilize the H12 from forming the presumed coactivator binding site, and as a result, the receptor must recruit a repressor peptide, which decreases the gene transcription. We also found the interesting “short-long” switch phenomenon of the agonist and inverse agonist, together with several other research groups [22,24]. For instance, Wang et al. observed that the ‘short’ inverse agonist, **P3**, switched to the agonist, **P4**, after an isopropyl group was added in the left side of **P3**, while agonist **P4** switched to a ‘longer’ inverse agonist when a longer group piperidin-ethanone was added in the left side of **P4** [24]. It is quite interesting that the agonist and inverse agonist bind to the same binding pocket of RORγt, but these molecules display the switchable function toward RORγt by adjusting the molecular length. What are the intrinsic factors that these molecules show the switchable function? In the present study, an agonist, **1**, and an inverse agonist, **2**, with a novel *N*-sulfonamide tetrahydroquinoline scaffold were designed, synthesized, and biologically evaluated, and were used as molecular tools to explore the possible answer to this interesting question. The switchable mechanism of the ‘short-long’ agonist and inverse agonist was elucidated at the molecular level by integrating computational techniques, including molecular docking and MD simulations. 

The MD simulations of the apo RORγt, agonist **1**-RORγt, and inverse agonist **2**-RORγt systems showed that, for all of the systems, the temperature, mass density, and volume are relatively stable after 2 ns along the MD trajectory. About then, the fluctuation scale became much smaller for all root mean square deviations (RMSD) of the backbone atoms of the protein, and the potential energy curve of each of the three simulation systems (Appendix A) indicates that the simulated system was well behaved thereafter.

### 3.1. Compound Design

Compound **GT** is one in the tertiary sulfonamide series, which were reported by scientists from Genentech (**GT** structure was shown in Figure 3). **GT** is identified as a potent RORγt inverse agonist with impressive selectivity over RORα and RORβ^4^. As a good starting point, we took **GT** as our lead compound and set a moderate structure modification, finally, we got some new RORγt compounds. Among these compounds, compound **1** and compound **2** caused our attention for their interesting bioactivity.

### 3.2. Biologically Evaluated RORγt Agonist ***1*** and Inverse Agonist ***2***

As shown in Figure 3 and Table 1, compound **1** was identified as an agonist of RORγt with an EC_50_ value of 3.7 μM (max. act.: 78%), while compound **2** was found as an inverse agonist with an IC_50_ value of 2.0 μM (max. inh.: −61%). We further evaluated the agonist, **1**, and inverse agonist, **2**, in the mouse Th17 cell differentiation assay. Agonist **1** showed some activity on enhancing the transcription of IL-17. Inverse agonist **2** showed some ability in inhibiting IL-17 transcription. Then, agonist **1** and inverse agonist **2** were used as tool compounds to explore why a minor change on the structure of some RORγt ligands led to a distinct function.

### 3.3. The Binding Modes of RORγt Agonist ***1*** and Inverse Agonist ***2***

The agonistic and inverse agonistic conformations of RORγt were extracted from the trajectory of MD simulations, and are shown in Figure 4. In terms of agonist-bound RORγt, the phenyl head of agonist **1** lays into the gap between H11, H3, and H12 that forms face-to-edge π-π hydrophobic interactions with His479 in H11 and Trp317 in H3. Such an interaction adjusts Trp317 to be in *gauche* conformation, thus, a large hydrophobic network was constructed by His479, Tyr502, Phe506, Trp317, and Phe486. In addition, His479 is hydrogen bonded with Tyr502 at H12. These strong hydrophobic and hydrogen bond interactions stabilize the H12. The acetyl moiety of agonist **1** forms a hydrogen bond with the sidechain of Gln286 (2.80 Å), and such inter-molecular interaction adjusts the orientation of agonist **1** at the active site of RORγt. In terms of inverse agonist-bound RORγt-LBD, the phenyl head of the longer inverse agonist **2** inserts deeply into the binding site, and it forms a face-to-edge π-π interaction with His479, but not with Trp317, making its side chain at *trans* conformation. As a result, the above described hydrophobic network involving Trp317 of H3 and Phe486 of H11 could not be formed. Due to the lack of stable hydrogen bond and hydrophobic interactions between H11 and H12, the H12 unwinds and becomes much more flexible. This could be the reason why the H12 became a loop at the crystal structure of RORγt bound with an inverse agonist (PDB entry as 5APK). 

### 3.4. Correlation between the Stability of H12 and Hydrophobic Interaction among H11, H11′, and H12

It is noteworthy that there is a hydrophobic network formed by Phe506 (from H12), Tyr502 (from H12), His479 (from H11), Phe486 (from H11), Trp317 (from H3), Trp314 (from H3), and His490 (from H11′) in the active site of apo RORγt, it is an important driving force to stabilize H12. In this hydrophobic network, Trp317 is located in the middle, and it connects two small hydrophobic clusters: WHYF and WWFH (as shown in Figure 5).

In terms of the agonist **1**-RORγt system, the phenyl group of agonist **1** inserts into the gap between Try317 and His479. Such an insertion helps to construct a larger hydrophobic network, and such tight interactions further help to drive H12 close to H11. 

In terms of the inverse agonist **2**-RORγt system, the benzyl group of the little longer inverse agonist **2** forms π-π interactions with His479, but it pushes Trp317 away, and such a push breaks the hydrophobic network around H11, H11′, and H12. The lack of hydrophobic interaction together with the lack of hydrogen bond between H11 and H12 destabilizes H12, and H12 has an obvious despiralization tendency. The crystal structures of solved inverse bound RORγt (PDB ID: 4QM0, 5APK, 5M96, 6CN6, etc.) show that their secondary structures of H12 could not be determined due to its larger flexibility. 

### 3.5. The Key Role of Trp317 in Agonism and Inverse Agonism of RORγt

As discussed above, the agonist can strengthen the hydrophobic network, but the inverse agonist breaks such a network. What is the intrinsic reason that the agonist and inverse agonist can bring about such different conformational states, and therefore reverse the gene transcription function of the receptor? The MD simulations for the apo RORγt, agonist-bound RORγt, and inverse agonist-bound RORγt identifies a key residue, Trp317, that plays an important role in adjusting the hydrophobic network. MD simulations identified two conformational states of Trp317 by monitoring χ_1_: It exists in *gauche*- conformation upon agonist binding and in *trans* conformation upon inverse agonist binding, while it interconverted from the *gauche* to *trans* conformational state in the RORγt system (as shown in Appendix A). 

In the agonist **1**-RORγt system, the phenyl head of agonist **1** inserts into the gap between His479 and Trp317, and it makes Trp317 keep its conformation to be the *gauche* state, which could stabilize the WWFH cluster (as shown in Figure 6A). The phenyl group of the agonist forms π-π interactions with His479 and Trp317, and thus Trp317 firmly connects the HYF cluster and WWFH cluster, so that a large hydrophobic network among H11, H11′, and H12 was soundly constructed. The observed *gauche*- conformation state of Trp317 is consistent with the conformation state in the agonists-bound RORγt’s crystal structures (PDB ID: 4NIE, 4XT9, 5APH, 5IZ0). Sequentially, it stabilizes H11′ and H12. Finally, the stable H12 can help the receptor recruit the co-activator, and therefore the RORγt is activated. This observation is verified by the secondary structure analysis through using the DSSP program [29]. As shown in Figure 6 and Appendix A, H11′ and H12 keep in the intact helix structure. 

On the contrary, in terms of the inverse agonist system, the longer phenyl head of inverse agonist **2** deeply inserts into the gap between His479 and Trp317, thus it could not afford a proper orientation for His479 to form a hydrogen bond interaction with Tyr502. The deeply inserted phenyl group of inverse agonist **2** could not form a π-π interaction with Trp317, and, instead, it pushes away Trp317. This push from the bound inverse agonist **2** forced the Trp317 to adjust its conformation to the *trans* state. The *trans* Trp317 is away from the π-π cluster of His479 and the phenyl group of inverse agonist **2**. This change of Trp317 agrees with what has been reported in X-ray crystal structures of RORγt bound with inverse agonists (PDB ID: 5APK, 4QM0, 4ZOM, 4ZJR, etc.). As a result, the large hydrophobic network could not be formed at all. In this situation, H11 cannot interact very well with H11′ and H12. Due to the weakened interactions with H11, the H11′ and H12 unwind (pink-colored panel in Figure 6A). Especially, H12 is located in the C terminal, it becomes quite flexible, so that its structure could not be determined in the reported inverse agonist bound crystal structures [21,22,30]. 

In terms of the apo system, the secondary structure of H11′ was loose following the conformational switch of Trp317 (Appendix A). After ca. 23 ns, the side chain of Trp317 switches from *gauche* into the *trans* conformational state. Correspondingly, Phe486 adjusts the torsion angle of its side-chain; sequentially, Trp314 and His490 changed their conformations (orange-colored panel in Figure 6). As a result, the WWFH cluster network formed by H11, H11′, and H3 was broken and disintegrated into a small hydrophobic cluster in H11 and H3. Besides, Trp314 and His 490 are located in the edge of H3 and H11′, respectively, and their flexible conformational changes destabilize the secondary structure of H11′, which is observed by the snapshots from 23 to 70 ns (top-right panel in Figure 6B), and this finally makes H11′ coil. H11′ works as a hinge to adjust the despiralization of H12 and makes H12 move away from H11. 

In the whole activation process, as described above, residue Trp317 is critical for the hydrophobic network among H11, H11′, and H12. The switchable conformational state of Trp317 adjusts the stability of H11′ by constructing the hydrophobic network. The ‘short-long’ agonist or inverse agonist triggers Trp317 to adopt the *gauche* or *trans* conformational state, respectively, and, therefore, the ‘short’ agonist **1** activates RORγt, but the ‘long’ inverse agonist represses its activation. Based on the key role of residue Trp317 as observed in our MD simulations, we can predict what a given compound will possibly be. If it is able to keep the Trp317 side chain in the *gauche* state, it should be an agonist. If the Trp317 side chain turns into the *trans* conformational state, the compound will be an inverse agonist.

## 4. The Molecular Mechanism of Action of RORγt Agonist and Inverse Agonist

In summary, when agonist **1** binds to RORγt (as shown at the left panel of Figure 7), Trp317 stays in the *gauche* conformational state, and it helps to form a large hydrophobic network among the hydrophobic residues from H11, H11′, and H12. The binding of the agonist also stabilizes the hydrogen bond, His479-Tyr502, between H11 and H12. These strong intra-molecular interactions of H11 with H11′ and H12 do stabilize the secondary structure of H11′ and H12. The stabilized H12 helps to recruit the coactivator and activate RORγt. Finally, the activated RORγt enhances the gene transcription. When inverse agonist **2** binds to RORγt (as shown at the right panel of Figure 7), residue Trp317 changes to the *trans* conformational state. The bound inverse agonist at the binding site, and the conformational change of the Trp317 side chain break down the possible hydrogen bond of His479 with Tyr502, and destroys the hydrophobic network among H11, H11′, and H12. Due to the lack of stable interactions between H11, H11′, and H12, the H12 unwinds and moves away from H11. In this process, the despiralization of H11′ acts as a hinge to adjust the inverse agonist induced conformational state of H12. Finally, the inverse agonist represses the activation of the RORγt (as shown in Appendix A). 

## 5. Conclusions

In the present study, we designed and synthesized an agonist and an inverse agonist of RORγt with the same scaffold. The agonism and inverse agonism of RORγt induced by the ‘short-long’ agonist and inverse agonist were firstly elucidated by integrating a series of computational techniques, including molecular docking and MD simulations. As we found, H11′ of RORγt acts as a hinge to coordinate the conformation of H12. The hydrogen bond, His479-Try502, and hydrophobic network around the H11, H11′, and H12 are constructed by the bound agonist, but partially destroyed by the binding of the inverse agonist. Based on the results of our MD simulations, we found that Trp317 played a critical role in the activation process of RORγt.

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
