# Peer review of "Molecular Mechanism of Action of RORγt Agonists and Inverse Agonists: Insights from Molecular Dynamics Simulation"

_molecules, 2018, doi:10.3390/molecules23123181_

Round 1
Reviewer 1 Report
In this study the investigators examine two new RORgt ligands. They show using FRET assay that one of them acts as an agonist and the other as inverse agonist. They further study the interaction of the ligand with RORgt protein using molecular dynamics simulations. They conclude that the agonist stabilizes H12 thereby promoting the interaction with co-activators, whereas the inverse agonist does the opposite. They conclude that Trp317 plays a crucial role in the activation of RORgt. The identification of the two novel RORgt ligands is interesting and overall performed well.
1. Investigators should include at least one or two additional assays to analyze the activity of the new RORgt ligands such as examining their effect on the activation of the IL17 promoter, a direct target gene of RORgt, or effect on IL17 expression in Th17 cells.
2. Do the investigators have any information of the effect of these novel ligands on other nuclear receptors? At least their effect on RORa should be examined and included.
3. What does the simulation with RORa tells you? Does it indicate that they also interact with RORa or that their interaction is very weak?
Author Response
Response to Reviewer 1 Comments Point 1: Investigators should include at least one or two additional assays to analyze the activity of the new RORgt ligands such as examining their effect on the activation of the IL17 promoter, a direct target gene of RORgt, or effect on IL17 expression in Th17 cells. Response 1: We have performed additional experiment of mouse Th17 differentiation assay, and provided details on Line 93-95 (on Page 4) and in the Supporting Information. The new results are summarized as Table 1 from Line 171 to Line 177 (on Page 7). Point 2: Do the investigators have any information of the effect of these novel ligands on other nuclear receptors? At least their effect on ROR should be examined and included. Response 2: This is a good question, and is worth further studies to explore the selectivity of our new compounds toward RORγt over ROR. Our compounds were designed based on the sulfonamide scaffold compounds which were identified as selective RORγt inverse agonists (Bioorganic & Medicinal Chemistry Letters 2014, 24, 5769–5776). We are devoted to develop the ROR assay. However, such assay is not available at this time. Based on our experience on the design and develop of RORγt ligands, we decided to first test the new compounds on RORγt, and focused the current study on the agonistic and inverse agonistic mechanisms toward RORγt, and we will report new results of the compounds’ selectivity in the near future. Point 3: What does the simulation with ROR tells you? Does it indicate that they also interact with RORa or that their interaction is very weak? Response 3: Based on the initial observation of the X-ray structure of ROR, for example PDB ID: 4S15 (Cell Matbolism 2015, 21, 286-298), we docked our compounds into ROR. Our test dock on the ROR showed that agonist 1 and inverse agonist 2 can’t fit well into the ligand binding site in RORstructure.

Reviewer 2 Report
In the manuscript entitled, “Molecular Mechanism of Action of RORgt Agonists and Inverse Agonists: Insights from Molecular Dynamics Simulation,” Sun et al., discover two novel modulators of ROR. The group has synthesized the two compounds, which are very similar in chemical structure. However, when tested in a FRET co-activator recruitment assay they display very different activating properties. One is classified as an agonist (nearly a full agonist) and one is classified as an inverse agonist. Both demonstrate low micromolar IC50s. Given the similarity in structure, the authors carried out docking and molecular dynamics simulations to shed light on the molecular mechanisms and structural differences between ROR agonists and inverse agonists. This paper is timely as compounds that target this receptor are importantly in drug discovery. The methods are sound and the interpretations are consistent with the data. I recommend publication with minor changes:
-The English language/grammar is very poor, detracting from the quality of the manuscript. Perhaps the journal could point them in the way of a quality language editor.
-The FRET assay measures co-activator recruitment. This is a nice experiment. However, in the grand scheme of things it is incomplete. Many nuclear receptor ligands recruit co-regulators to different degrees. It would be critical to do a classical transactivation assay for this paper to also show transactivation potential and potency.
-Hydrogen bond distance labels in the figures would be helpful to the reader.
-The structural figures in general could be just a tab bigger.
-Line 236, the analogy to conformations, “like domino” seems confusing and I wasn’t able to understand it.
-It is common to have MD videos in the supplement
Author Response
Response to Reviewer 2 Comments
Point 1: The English language/grammar is very poor, detracting from the quality of the manuscript. Perhaps the journal could point them in the way of a quality language editor.
Response 1: Based on the reviewer’s comments, we have asked the help from a professional scientist in the United States of American to rewrite and polish the whole manuscript, and we hope this revised version is much better, thanks.
Point 2: The FRET assay measures co-activator recruitment. This is a nice experiment. However, in the grand scheme of things it is incomplete. Many nuclear receptor ligands recruit co-regulators to different degrees. It would be critical to do a classical transactivation assay for this paper to also show transactivation potential and potency.
Response 2: we have performed additional experiment of mouse Th17 differentiation assay, and provided details on Line 93-95 (on Page 4) and in the Supporting Information. The new results are summarized as Table 1 from Line 171 to Line 177 (on Page 7).
Point 3: Hydrogen bond distance labels in the figures would be helpful to the reader.
Response 3: Thank, it has been added in Figures 4B and Figure 4E.
Point 4: The structural figures in general could be just a tab bigger.
Response 4: Thanks, they are adjusted to be a bigger tab.
Point 5: Line 236, the analogy to conformations, “like domino” seems confusing and I wasn’t able to understand it.
Response 5: It is corrected, thanks.
Point 6: It is common to have MD videos in the supplement
Response 6: We provided four MD videos in the Support Information. These are the videos showing conformational change of apo RORγt-LDB system, the system of RORγt bound with the agonist 1, the system of RORγt bound with the inverse agonist 2, and the MOA of RORγt agonism and inverse agonism.
Reviewer 3 Report
This ms studied the interaction of ligand and one ROR, specifically investigated how agonist and inverse agonist conformation is related to W317’s sidechain conformation. Overall the ms is clearly written and very interesting to read. Here are some of the things the authors should consider.
Authors stated 67 ROR gamma t crystal structures already exist and all are not apo form. What is the sidechain structure of W317 in those cases? Are they consistent with author’s conclusion?
With these many existing ligand-bound structures, what was the rationale of using docking software to obtain a new ligand structure for this study. Normally docking is used to explore variety of possibilities. Without another corroboration or validation, they are not a starting point of a long-time simulation to further investigate the mechanism. Many situations it was the only choice, but it does not appear to be this case. Is there a better way to model the initial ligand structure or the docking result corroborate with some known feature? Also, docking can be quite accurate in general, but in the case of nuclear receptor families, can author comment on how secure/experienced it is for this class of receptors.
The results are focused on W317 conformation plays the central role. How does the authors narrow it down from all possibilities to this one residue? Also what does it mean for this identification/mechanism? Is it really the causality or just one of many correlations? Can authors now based this discovery predict any new ligand would be agonist or inverse agonist based on this finding?
Minor issues on the presentation style: for example, the authors called the two ligands simply “1” and “2” in the abstract and introduce them also quite late in the main text. Shouldn’t them be given a name or at least some sort of chemical identities or something? Even when introduced in the abstract as 1 and 2, 1 is called “ligand” and 2 is called “compound.” These subtle inconsistencies make the paper confusing for readers at time.
Author Response
Response to Reviewer 3 Comments
Point 1: Authors stated 67 ROR gamma t crystal structures already exist and all are not apo form. What is the sidechain structure of W317 in those cases? Are they consistent with author’s conclusion?
Response 1: Thanks for this comment. We added such information at Line 239-241, Line 251-252 (on Page 9 and Page 10). Briefly, in terms of agonists-bound RORγt’s ctystal structures (PDB ID: 4NIE, 4XT9, 5APH, 5IZ0), c1 of Trp317 is in gauche- conformation. In terms of inverse agonists-bound RORγt’s ones (PDB ID:5APK, 4QM0, 4ZOM, 4ZJR, etc.), c1 of Trp317 is in trans conformation. Our observation about the conformational states of Trp317 is consistent with these in the published X-ray crystal structures.
Point 2: With these many existing ligand-bound structures, what was the rationale of using docking software to obtain a new ligand structure for this study. Normally docking is used to explore variety of possibilities. Without another corroboration or validation, they are not a starting point of a long-time simulation to further investigate the mechanism. Many situations it was the only choice, but it does not appear to be this case. Is there a better way to model the initial ligand structure or the docking result corroborate with some known feature? Also, docking can be quite accurate in general, but in the case of nuclear receptor families, can author comment on how secure/experienced it is for this class of receptors.
Response 2: Thanks for the comments. We provided information at Line 107-109 (on Page 4) of how the binding site was initially defined. Actually, we did test docking operations to explore how reliable of the docking program GlideXP for the study of agonist 1 and inverse agonist 2. We selected 10 crystal structures of RORγt complexes, including the ones in which the ligand’s shape is a kind of similar to our new compounds. The predicted binding energies and docking frequencies were used as criteria to select the final docking poses. The results of test docking show that program GlideXP can reproduce the binding modes as those in the crystal structures, and the positional root-mean square deviations (RMSD) of docked ligands for all systems are less than 1.5 Å. Indeed, we performed docking very carefully for our molecules, and use the same criteria in selecting the final docking poses.
Point 3: The results are focused on W317 conformation plays the central role. How does the authors narrow it down from all possibilities to this one residue? Also what does it mean for this identification/mechanism? Is it really the causality or just one of many correlations? Can authors now based this discovery predict any new ligand would be agonist or inverse agonist based on this finding?
Response 3: Good comments. Based on our knowledge, this is the first time to investigate the mechanisms of agonist and inverse agonist of RORγt at atomic level. We carefully analyzed all hydrogen bond networks and hydrophobic clusters for the apo, agonistic and inverse agonistic systems. We found that Trp317 is located in the center of a big hydrophobic cluster (Phe506, Tyr502, His479, Phe486, Trp317, Trp314 and His490), and this big hydrophobic cluster can affect the interactions of the functional regions H3, H11, H11’ and H12. By monitoring the c1 of Trp317, an interesting phenomenon was observed: Trp317 exists in gauche- conformation upon agonist binding and in trans conformation upon inverse agonist binding. The gauche- or trans conformation of Trp317 push the conformational adjustments of Phe486, Trp314 and His490, and so, the RORγt is agonized or inversely agonized. Based on the results of our MD simulations, we can predict that a given compound will be an agonist if it is able to keep the Trp317 side chain in the gauche- state, and it will be an inverse agonist when the Trp317 side chain turns to the trans conformation. We added this information at Line 272-275 (on Page 12)
Point 4: Minor issues on the presentation style: for example, the authors called the two ligands simply “1” and “2” in the abstract and introduce them also quite late in the main text. Shouldn’t them be given a name or at least some sort of chemical identities or something? Even when introduced in the abstract as 1 and 2, 1 is called “ligand” and 2 is called “compound.” These subtle inconsistencies make the paper confusing for readers at time.
Response 4: Thanks, and we updated their names to be agonist 1 and inverse agonist 2 in the revised manuscript.
Round 2
Reviewer 1 Report
The revised paper has been improved: however, one important issue about RORg specificity was not clearly addressed.
1. Line 49: “to guild” should be “to guide”
2. Line 168: “in inhibition of IL-17’s transcription” change to “inhibiting IL-17 transcription”
3. Line 194: “hydrogen bonding” change to “hydrogen bond”
4. It would have been nice to know more about the specificity of the compounds particularly whether they interact with the closely related RORa. I made this comment previously but I do not see that this point is addressed in the paper. RORg specificity is an important issue that If the investigators have any info on this needs to be included.
Author Response
Point 1: Line 49: “to guild” should be “to guide”.
Response 1: It has been corrected in the revised manuscript.
Point 2: Line 168: “in inhibition of IL-17’s transcription” change to “inhibiting IL-17 transcription”.
Response 2: It has been corrected in the revised manuscript.
Point 3: Line 194: “hydrogen bonding” change to “hydrogen bond”.
Response 3: It has been corrected in the revised manuscript.
Point 4: It would have been nice to know more about the specificity of the compounds particularly whether they interact with the closely related RORa. I made this comment previously but I do not see that this point is addressed in the paper. RORg specificity is an important issue that If the investigators have any info on this needs to be included.
Response 4: We appreciate the reviewer highlighted again. We agree this is important, and we do think the specificity over RORa is not critical to our conclusion that “compound 1 is the agonist, and compound 2 is the inverse agonist of RORgt”. The insights into the molecular mechanism of action (MOA) from our MD simulation do support our results of the FRET and the newly added TH17 differentiation assays. It is not common to perform such specificity test (for example, ACS. Med. Chem. Lett. 2018, 9, 120-124; European Journal of Medicinal Chemistry 2016, 116, 13-26; Bioorganic & Medicinal Chemistry Letters 2017, 27, 5277–5283; etc.). Actually, it will take large efforts to build and perform such test, so we are sorry that we do not have such results for the time being.
Reviewer 2 Report
Figure 3 needs a standard ROR control. There should be a transactivation assay (EC50) to complement the co-activator recruitment assay (not an IC50 of differentiation).
Author Response
Point 1: Figure 3 needs a standard ROR control. There should be a transactivation assay (EC50) to complement the co-activator recruitment assay (not an IC50 of differentiation).
Response 1: Thanks for this point. In the current study, we used the RORγt inverse agonist GT from Genentech (Bioorganic & Medicinal Chemistry Letters 2014, 24, 5769–5776) as a positive control and the data has been added in Table 1 and Figure 3. We performed the FRET assay and Th17 differentiation assay to test the EC50 of RORγt agonist, and the IC50 of RORγt inverse agonist. Th17 cell differentiation assay tests the effect of RORγt agonist and inverse agonist on IL-17 production by CD4+ T cells under conditions which favor Th17 cell differentiation. Based on the available literature, we found that Th17 differentiation assay could complement the co-activator recruitment assay (for example, ACS Med. Chem. Lett. 2018, 9, 120−124; ACS Chem. Biol. 2016, 11, 1012−1018; J. Med. Chem. 2015, 58, 5308−5322; etc.). In addition, we corrected the typo in Table 1 (on Page 6, from Line 180), that is, the EC50 value of agonist 1 obtained by FRET and Th17 cell differentiation assay.